# Patient-Related Factors of Medialization Laryngoplasty with Autologous Thyroid Cartilage

**DOI:** 10.3390/healthcare8040521

**Published:** 2020-11-30

**Authors:** Yao-Te Tsai, Ming-Shao Tsai, Geng-He Chang, Li-Ang Lee, Ming-Yu Yang, Yao-Hsu Yang, Chin-Yuan Wu, Cheng-Ming Hsu

**Affiliations:** 1Department of Otolaryngology-Head and Neck Surgery, Chang Gung Memorial Hospital, Chiayi 613016, Taiwan; yaote1215@gmail.com (Y.-T.T.); b87401061@cgmh.org.tw (M.-S.T.); a9244@cgmh.org.tw (G.-H.C.); 2Graduate Institute of Clinical Medical Sciences, College of Medicine, Chang Gung University, Taoyuan 333323, Taiwan; 3Department of Otolaryngology-Head and Neck Surgery, Chang Gung Memorial Hospital, Linkou 333423, Taiwan; 5738@cgmh.org.tw; 4Faculty of Medicine, College of Medicine, Chang Gung University, Taoyuan 333323, Taiwan; yangmy@mail.cgu.edu.tw (M.-Y.Y.); smbepig@cgmh.org.tw (C.-Y.W.); 5Department of Traditional Chinese Medicine, Chang Gung Memorial Hospital, Chiayi 613016, Taiwan; r95841012@cgmh.org.tw

**Keywords:** glottal closure insufficiency, voice analysis, sex, age, body mass index

## Abstract

(1) Background: Medialization laryngoplasty with autologous thyroid cartilage (MLATC) is a surgical treatment for glottal closure insufficiency (GCI) resulted from unilateral vocal fold paralysis/paresis (UVFP) and vocal fold atrophy. We aimed to survey the influence of patient-related factors on the outcomes after MLATC. (2) Methods: The study enrolled 35 patients with GCI who underwent MLATC. Patient voice data were recorded before and after MLATC by using multiple acoustic parameters and subjective assessment in a computerized speech laboratory. GCI patients were characterized into subgroups based on three factors: age, ≥60 vs. <60 years; sex, men vs. women; and BMI, ≥24 vs. <24. (3) Results: When the subgroups were compared, men did not have better results after surgery than women. Patients ages < 60 years did not exhibit any significantly different outcome compared with those aged ≥ 60 years. Patients with BMI ≥ 24 did not have any significantly different outcome compared with those with BMI < 24. The subgroups of age, sex, and BMI had no significant difference in cumulative voice recovery and summation of GRBAS (G = grade, R = roughness, B = breathiness, A = asthenia, and S = strain). (4) Conclusions: MLATC is a good alternative surgery with long-term improvement in GCI patients. There is no evidence that age, sex, or BMI affect the functional outcome.

## 1. Introduction

Glottal closure insufficiency (GCI) is a common cause of voice disorders and results from the uncompleted closure of the vocal folds [1,2] due to mucosal lesions or vocal mobility [3]. Neuromuscular disorders and joint dislocation can cause unilateral or bilateral vocal fold immobilities, e.g., unilateral vocal fold paralysis (UVFP), and the mucosal lesions may include vocal defects, scarring, sulcus vocalis, and atrophy [4,5]. Severe GCI may present with dysphagia, dysphonia, and aspiration, and can lead to pneumonia and even death [6]. The standard treatments for GCI include injection laryngoplasty, medialization laryngoplasty, arytenoid adduction, and laryngeal reinnervation [7,8]. Although the office-based injection laryngoplasty (IL) can be easily performed under local anesthesia without gross wounds, it is less effective for patients with severe posterior GCI and has inferior long-term effects because of the absorption of the injected materials [9,10].

Medialization laryngoplasty (ML), when an implant is inserted through a lateral window in the thyroid cartilage to medialize the vocal folds, is an effective surgical procedure for treating glottal insufficiency, and a Montgomery thyroplasty Implant System with a Silastic block has been developed to treat unilateral vocal fold paralysis through medialization of the paralyzed cord [11]. Autologous cartilage is an alternative graft material because of its high biocompatibility, low absorption rate, and negligible tissue reaction. As demonstrated in our previous study, modified medialization laryngoplasty through autologous thyroid cartilage (MLATC) is a practical surgical treatment for UVFP [2,12], which can reduce the glottal gap, medialize the vocal fold, and mitigate GCI without silastic or foreign implants. However, the long-term results of MLATC remain unclear because of limited cases and the lack of relevant studies; therefore, the primary endpoint of this study was to investigate the long-term results of MLATC.

In addition, there may be some factors that will affect the outcome of MLATC. In the elderly, the collagen and elastic fibers within the lamina propria of larynx undergo changes, and the mucous glands and membranes may atrophy as well; therefore, the outcome of MLATC may be affected by aging [13]. The different thyroid cartilage size, hardness, and calcification degrees between men and women patients may also influence the result of MLATC. Moreover, patients with less paraglottic fat and body mass index (BMI) may have different treatment outcomes. Increasing phonosurgeons are choosing MLATC because of its convenience and effectiveness, however, whether patient factors, e.g., sex, age, and BMI, are indicators of long-term results of MLATC has not been reported in the literature. Clearly, a deeper understanding of the prognosticators in patients undergoing MLATC is required; therefore, the second endpoint of this study was to investigate the influence of age, sex, and BMI on the long-term voice results after MLATC.

## 2. Materials and Methods

This study was conducted in accordance with the guidelines and regulations of Institutional Review Board of Chang Gung Memorial Hospital that approved this study (No. 100–2757B). The review board waived the requirement for obtaining informed consent from the enrolled patients.

### 2.1. Human Subjects

Between August 2012 and March 2019, we performed a retrospective cohort study and evaluated 35 patients newly diagnosed as having GCI who received Isshiki type I thyroplasty with autologous thyroid cartilage implants at the Department of Otorhinolaryngology, Chiayi Chang Gung Memorial Hospital. Prior to surgery, all patients experienced moderate to severe dysphonia for more than 6 months. The eligibility criteria were as follows: (1) age more than 18 years; (2) diagnosis of GCI confirmed through laryngostroboscopy; (3) treatment with MLATC at our hospital; and (4) complete analyses of preoperative and postoperative vocal functions with acoustic analysis and perceptual assessment. The exclusion criteria were as follows: (1) inoperable cancer or contraindication for surgery; (2) malignancy or distant metastasis at presentation; and (3) missing preoperative or follow-up data. All patients underwent standard routine preoperative assessment according to institutional guidelines, and sex, age, BMI, UVFP side, cause of UVFP, preoperative severity of symptoms, and postoperative voice outcomes were documented from medical record review. After discharge, the patients underwent outpatient clinic follow-up visits every 3 months for the first year and long-term follow-up at least 12 months. At each follow-up visit, perceptual assessment and laryngostroboscopy with acoustic analysis were performed.

Under regional anesthesia with intravenous sedation, a 4-cm horizontal incision was made on the lesion side and the underlying entire thyroid cartilage was exposed after the strap muscles were dissected laterally. Electrocautery was used to mark the third quarter of the midline of the thyroid cartilage. The anterior edge of the thyroid cartilage window was marked parallel and 7 mm laterally to the first mark for men and women, respectively. The width of the window is usually 3–5 mm, and the length of the thyroid cartilage window is usually 15–18 mm, with an inferior border 5 mm superior to the lower border of the thyroid cartilage. The delineated window was cut with a number-15 blade, and the cartilage in the window was removed using a curved clamp.

We used a mucosal elevator to free the inner perichondrium from the cartilage and create a space for the insertion of a cartilage implant. We tested patients’ voice by pressing the soft tissue with the mucosal elevator through the window before taking a graft. While the patients phonated a long “e” sound, the surgeon pressed the vocal fold toward the midline. This intraoperative testing determined the proper size of the graft.

On the same side as the laryngeal prominence, the upper portion of the thyroid cartilage was excised and trimmed into a lunar-shaped graft, which was then inserted into the window at an oblique angle. The curved side was oriented toward the paralyzed vocal fold. The remaining graft was pressed into the window by using a mucosal elevator and locked tightly without suturing. In most surgeries, we do not need to adjust the position of cartilage. Fine vocal adjustment was only needed to downsize the cartilage, because cartilage slightly larger than its ideal size was used. Simultaneous arytenoid adduction was performed to achieve optimal voice results. Simultaneous arytenoid adduction was performed for patients who still had a glottal gap after graft implantation.

To increase stability, the height of the window was required to closely fit the implant thickness, because cartilage grafts are vulnerable to dislocation [2,12]. A closed wound drain was placed and then removed within 3 days. In most cases, postoperative antibiotics were not required. The perioperative and postoperative complications included compromising airway and postoperative hematoma or infection. Emergent tracheostomy is rarely needed but still possible.

### 2.2. Voice Parameters Analysis

To investigate the voice treatment outcomes after MLATC as well as the impact of patient factors, e.g., sex, BMI, and age, we compared the preoperative and postoperative voice parameters and performed subgroup analysis based on these patient factors. The vocal functions within 2 weeks before surgery and on the 3rd and 12th months after surgery were documented and analyzed using laryngeal strobovideoscopy with the KayPENTAX stroboscopy system (Model 9400) from KayPENTAX (Lincoln Park, NJ, USA). The acoustic parameters, including the average F0 (Hz), jitter, shimmer, and noise-to-harmonic ratio (NHR), were recorded automatically. The jitter represents the cycle variations of the fundamental frequency, and the shimmer represents the variability of the peak–to–peak amplitude in decibels [14,15]. NHR is a measure of the amount of additive noise in the voice signal used to evaluate a dysphonic voice [16]. These acoustic parameters were measured in a Computerized Speech Laboratory (core model CSL 4500, KayPentax, Lincoln Park, NJ, USA). The duration for which a parameter was recorded was the maximum phonation time (MPT), and a speaker’s MPT was recorded using a microphone as the they pronounced the vowel “a” for as long as possible after a deep breath [17]. Preoperative and postoperative perceptual analysis was performed using the GRBAS scoring system (G = grade, R = roughness, B = breathiness, A = asthenia, and S = strain; 0 = normal, 1 = mild, 2 = moderate, and 3 = severe). The GRBAS-sum is the summation of each GRBAS score. The rating was made by the speech pathologist and otolaryngologist separately on a short standard passage. The mean of two values recorded was used for analysis. If a difference of more than 2 points was noted between the two GRBAS scores, reevaluation was required. If still more than 2 points, the ratings were averaged.

A speech pathologist and an otolaryngologist (C.M.H.) analyzed the above-mentioned voice parameters in a double-blinded manner. In most developed countries, older adults are defined as those aged 65 years or older, and the United Nations (UN) has agreed that the cutoff age for categorizing older adults is 60 years, and a BMI more than 24 is currently classified as overweight [18]. Thereafter, the subgroup analysis based on sex (men vs. women), age (≥60 vs. <60), and BMI (≥24 vs. <24) was performed, and we compared the MPT and summation of GRBAS scores before and after surgery because they present the most frequently used voice outcome indicators [19].

### 2.3. Subjective Assessment

The Voice Handicap Index (VHI) is well-studied questionnaire for measuring the impact of dysphonia. VHI-10 and Chinese VHI-10 (VHI-C10) have been shown to be valid instruments for mandarin users.

### 2.4. Statistical Analysis

The data were analyzed using SPSS software (SPSS Inc., Chicago, IL, USA). The t test was used to analyze and compare the parameters before and after MLATC. Repeated-measures ANOVA was applied to compare the effects of operation, BMI, age and sex on the parameters. Voice recovery was defined as a significant improvement in the summary of the GRBAS and an improvement in MPT of more than 50%. Kaplan–Meier improvement curves were compared using a log-rank test.

## 3. Results

### 3.1. Voice Outcomes

The characteristics of all the 35 patients who underwent MLATC are presented in Table 1. The most common cause of GCI was UVFP (n = 31, 89%) followed by vocal atrophy (n = 4, 11%). Among those with UVFP, most paralysis occurred in left side vocal fold (n = 26, 84%), and the surgical sequela was the most frequent etiology (n = 24, 77%), followed by non-surgical cause (n = 5, 16%) and idiopathy (n = 2, 6%). The median age of patients was 58.2 ± 11.6 years old. The average postoperative follow-up duration was 42.5 months. All the postoperative data were obtained from postop at at least 3 months. A total of three cases were lost at follow up after 3 months. The data of 32 cases were obtained at 12 months. However, patients were followed as long as was possible, and the result still be recorded.

The acoustic analysis and subjective voice analysis are summarized in Figure 1 and Table 2. VHI-10, MPT, F0, Jitter, and NHR significantly improved after surgery (Figure 1a–e). Postoperative shimmer decreased without statistical significance (Figure 1f). All the differences of the GRBAS ratings by two raters were less than one. No reevaluation was needed. Pre- and post-operative perceptual assessments were conducted according to GRBAS scoring, which revealed a significant decrease in GRBAS-sum voice grade, and breathiness (Figure 1g and Table 2). The improvement of MPT lasts for a minimum of 1 year (Figure 1h). The axial view of head and neck computed tomography was obtained from four patients at 3 years after operation, and all images revealed that the implanted cartilage located ideally and medialized the paralyzed vocal fold (Figure 2).

### 3.2. Comparison of the Outcome of MLATC between Patients’ Aged ≥ 60 and Those Aged < 60

The surgical outcomes were affected by age. The patients were divided into two groups (age ≥ 60 and <60 years), and the results were analyzed. The acoustic analysis revealed a statistical difference in pre- and postoperative voice variables between patients aged ≥60 and those aged <60 (Table 3). The GRBAS-sum of the ≥60 age group was the same as that of the <60 age group (Figure 3a). No significant difference was observed between the different age subgroups for the MPT recovery (*p* = 0.29; Figure 3b).

### 3.3. Comparison of the Outcome of MLATC between Women and Men Patients

The surgical outcomes were affected by sex; therefore, the patients were divided into two groups (women and men), and the results were analyzed. The acoustic analysis revealed statistical differences between the two subgroups for some of the pre- and postoperative voice parameters (Table 4). A comparison between the two subgroups indicated no significant differences in terms of the GRBAS-sum (Table 4 and Figure 3c). No significant difference in the MPT recovery was observed between the women and men subgroups (*p* = 0.32; Figure 3d).

### 3.4. Comparison of the Outcome of MLATC between Patients with High and Low BMI

Patients with different BMIs may demonstrate different surgical outcomes. Overweight is defined as having a BMI over 24. Therefore, the patients were divided into two groups based on their BMI (BMI ≥ 24 and BMI < 24), and the outcomes of surgery were analyzed (Table 5). The acoustic analysis revealed a statistical difference between the two subgroups for some of the pre- and postoperative voice parameters (Table 5). A comparison of the two subgroups revealed no significant differences in terms of the GRBAS-sum (Table 5 and Figure 3e). Moreover, no significant difference in the MPT recovery was noted between the two BMI subgroups (*p* = 0.85; Figure 3f).

### 3.5. Repeated-Measures ANOVA for Analysis of Different Effect on Variables

Using multiple *t*-tests is not the best way to analyze these data, and correction factors are still needed. Hence, a repeated-measures ANOVA would be a better option. We analyzed the effect of operation, age, BMI and sex on the voice variable. The result was the same as previous multiple t-tests (Table 6).

## 4. Discussion

Autogenous cartilage is widely considered to be an ideal material for many types of grafts in facial reconstruction [20]. This material survives as living tissue, does not stimulate an immune response, and seldom undergoes resorption [21]. MLATC is an alternative approach for treating GCI; this approach yields significant postoperative improvements, both subjectively in terms of perceptual assessments and objectively in terms of voice parameters, even at long-term follow-up. According to the study of Sun Ryu, et al. the GRBAS scale (grade of hoarseness, roughness, breathiness, asthenia, and strain) values showed significant improvement at 6 months after the operation by using silicone blocks; these improvements continued up to 1 year and were maintained 5 years after the operation. The results of cartilage were as good as silicone blocks.

Posttreatment voice outcomes were influenced by age [13]. The elastin and collagen fibers within the lamina propria of the larynx undergo changes during the aging process. Moreover, thinning and atrophy of the mucous membranes and atrophy of the mucous glands are observed. In addition, younger patients have better pulmonary function and higher laryngeal phonation power. A comparison of two age subgroups of GCI patients (≥60 vs. <60 years) revealed that the patients aged ≥ 60 performed MPT or GRBAS-sum after receiving MLATC as well as those aged < 60.

Our study findings revealed no significant difference between men and women patients in terms of MPT or GRBAS-sum performance after receipt of MLATC. According to previous studies, the thyroid cartilage and larynx of men and women present several differences. Calcification and ossification are usually found in the thyroid cartilage of men. In general, the cartilage in men is larger and firmer [22]. When the cartilage of men is used as a graft in the larynx, higher strength and maintenance for medialization could be achieved; however, our results did not support this finding. Sex-specific outcomes must still be evaluated for the surgeon during preoperative counseling [23].

Our findings indicate that obese (BMI ≥ 24) patients performed shorter MPT and had higher GRBAS-sum scores after MLATC compared with patients who had a BMI < 24, but the result was not significant (*P* > 0.05, Table 5). Obese patients may have a higher fat component to their larynx, which may affect the outcome of medialization by graft. The paraglottic space is a fat-containing space located on either side of the larynx. Although the fat within the paraglottic space seems important for the vocal fold gap [24], obesity did not affect the outcome. BMI is not a critical factor in MLATC. However, the surgery is more difficult in obese patients with shorter necks, which could affect the outcome.

In a study by Farzal (2019) et al. no significant difference between sexes was noted in perceptual measures (GRBAS) for type I Gore-Tex thyroplasty for nonparalytic GCI [23], which is similar to our study finding. In injection laryngoplasty for glottal insufficiency, age and sex did not affect outcome. Patients with more severe disease appeared to exhibit greater improvement. Regardless of the treatment used for GCI, the results should not be affected by patient-related factors.

Aging does not affect the outcome of medialization laryngoplasty; older patients with vocal fold paralysis and vocal handicap can be successfully treated [25]. In another study, the BMI of patients with UVFP (BMI ≥ 24 vs. BMI < 24) did not affect voice quality following autologous fat injection laryngoplasty [26]. These results are similar with those of our MLATC results.

The results of repeated-measure ANOVA could confirm that there was no effect of BMI, age, and sex on the voice parameters. Patient-related factors had no effect on MLATC for treating GCI patient.

There are three limitations to this study. First, the sample size was small, with only 35 patients studied. Second, the outcome of cancer patients is affected by their underlying disease; almost 50% of GCI cases were caused by cancer or cancer surgery (48.5%, 17/35). The patient survival affects the follow-up time; however, the GCI cases are usually caused by poor cancer condition, which limited our long-term study. Third, the cause of GCI is not a normal distribution in this study. Further studies investigating MLATC are necessary.

## 5. Conclusions

MLATC is a suitable option for patients with GCI who prefer not to undergo injection laryngoplasty with temporal effects. This research suggests that patients with GCI who receive MLATC surgery could exhibit a significantly improvement in their voice outcome for a long time. The results confirmed that patient factors, including sex, age, and BMI, did not affect long-term outcomes. MLATC is a novel alternative surgery that can be performed in patients with GCI and yields long-term improvement.

## Figures and Tables

**Figure 1 healthcare-08-00521-f001:**
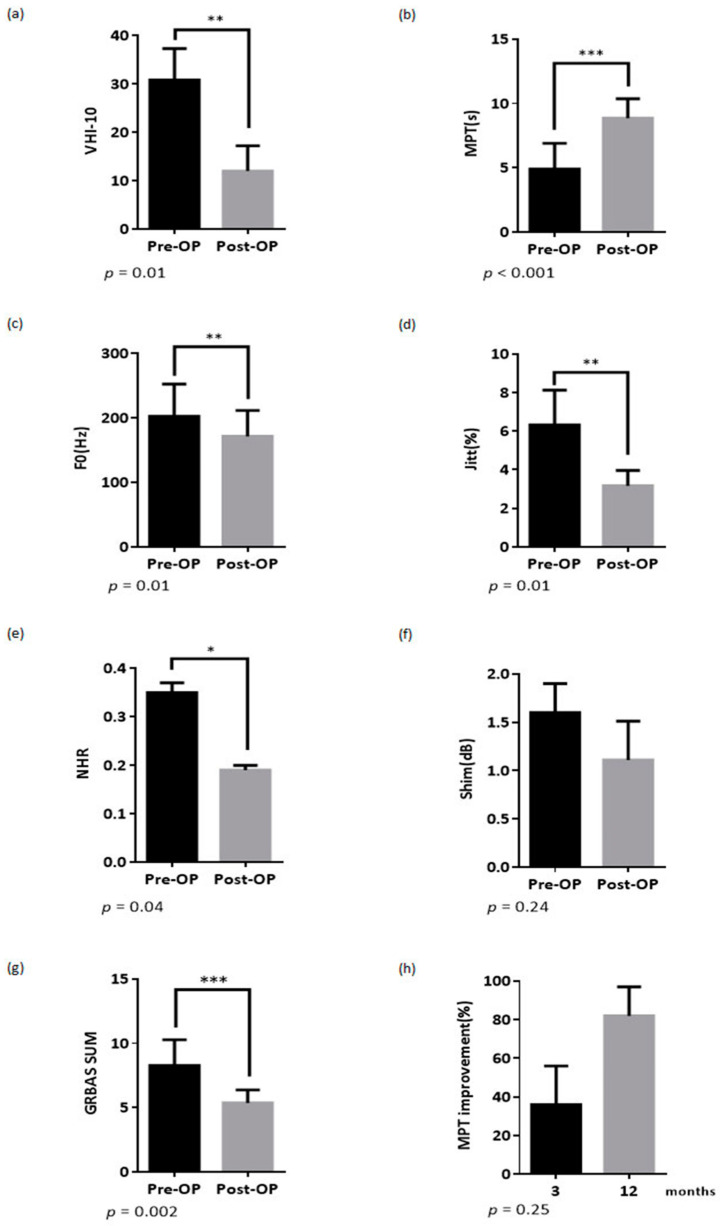
The pre-operative and post-operative acoustic analysis result of study patients (**a**) Pre- and postoperative Voice Handicap Index (VHI)-10 scores, (**b**) Pre- and postoperative maximum phonation times (MPT)s, (**c**) Pre- and postoperative F0, (**d**) Pre- and postoperative Jitt, (**e**) Pre- and postoperative noise-to-harmonic ratio (NHR), (**f**) Pre- and postoperative shimmer, (**g**) Pre- and postoperative GRBAS-sum (G = grade, R = roughness, B = breathiness, A = asthenia, and S = strain), (**h**) MPT improvement at 3 and 12 months. The * *p* < 0.05, ** *p* < 0.01 and *** *p* < 0.001 indicate the statistical significance of differences between pre-operation and post-operation.

**Figure 2 healthcare-08-00521-f002:**
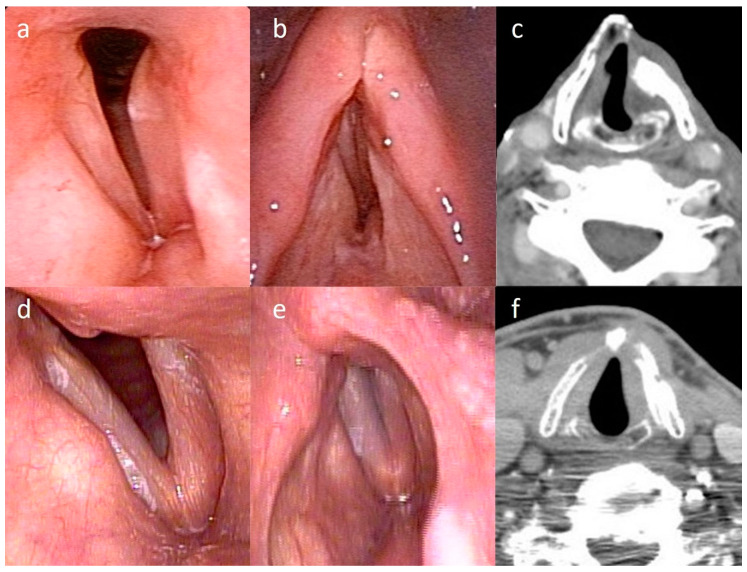
The images of laryngoscopy and axial view of computed tomography were obtained from 2 different cases’ larynx after operation for (**a**–**c**) 4 years and (**d**–**f**) 3 years, respectively, and all of them showed implanted cartilage still exists and medializes the left vocal fold.

**Figure 3 healthcare-08-00521-f003:**
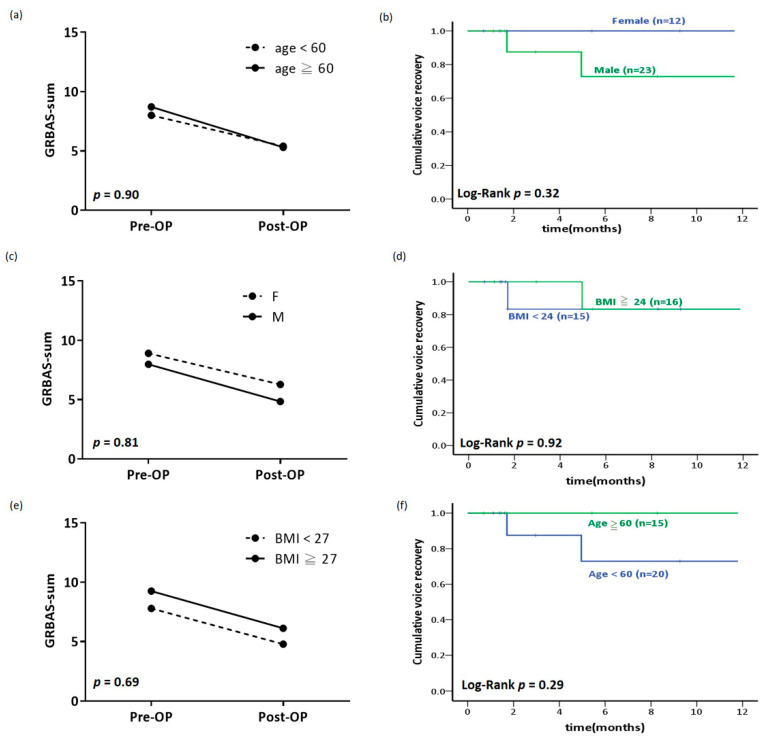
The change in grade, roughness, breathiness, asthenia, and strain-sum percentage by (**a**) age, (**c**) sex, and (**e**) body mass index before and after Medialization laryngoplasty with autologous thyroid cartilage (MLATC). The Kaplan-Meier plot of cumulative voice recovery of (**b**) age, (**d**) sex, and (**f**) body mass index after MLATC.

**Table 1 healthcare-08-00521-t001:** Patient characteristics.

Variable	Overall	Range
Age (years)	58.16 ± 11.61	33–85 y/o
age ≥ 60, n (%)	15 (43)	data
age < 60, n (%)	20 (57)	
Sex		
Men, n (%)	23 (56)	
Women, n (%)	12 (34)	
Total following time (months)	37.96 ± 35.07	3–113.88
Causes		
Paralysis, n (%)	31 (89)	
Atrophy, n (%)	4 (11)	
Cause of paralysis		
Surgical sequala, n (%)	24 (77)	
Non-surgical reason, n (%)	5 (16)	
Idiopathy, n (%)	2 (6)	
Site of paralysis		
Left, n (%)	26 (84)	
Right, n (%)	4 (13)	
Bilateral, n (%)	1 (3)	
BMI (mean)	24.49 ± 3.80	18.49–32.69
BMI ≥ 24, n (%)	16 (43)	
BMI < 24, n (%)	15 (46)	

Data are expressed as mean ± standard deviation or number (%).

**Table 2 healthcare-08-00521-t002:** Comparison of GRBAS between pre-OP and post-OP.

Variables	Pre-Operation	Post-Operation	*p*-Value
Grade	2.43 ± 0.77	1.63 ± 0.89	0.004
Roughness	1.74 ± 0.89	1.33 ± 0.80	0.09
Breathiness	2.34 ± 0.86	1.07 ± 0.94	0.004
Asthenia	1.28 ± 0.92	0.73 ± 0.94	0.118
Strain	0.97 ± 1.05	0.60 ± 0.67	0.11

Data are expressed as mean ± standard deviation. Results for which *p* values indicated significance are marked in bold.

**Table 3 healthcare-08-00521-t003:** Comparison of all parameters between the age ≥ 60 and age < 60 subgroups.

Variable	Age ≥ 60 (n = 15)		Age < 60 (n = 20)		Age ≥ 60 vs. Age < 60	
	Pre-Operation	Post-Operation	Pre-Operation	Post-Operation	*p*-Value ^1^	*p*-Value ^2^
VHI-10	**27.25 ± 4.27**	**11.67 ± 6.35 ^3^**	**38.00 ± 1.41**	**12.25 ± 8.52 ^3^**	**0.062**	0.951
MPT (sec)	**3.88 ± 2.00**	**9.43 ± 4.79 ^3^**	**5.26 ± 4.22**	**9.49 ± 5.82 ^3^**	0.343	0.978
F0 (Hz)	**184.27 ± 54.83**	**159.33 ± 36.34 ^3^**	216.23 ± 94.97	178.65 ± 63.04	0.289	0.342
Jitter	6.31 ± 4.24	3.51 ± 2.42	**6.32 ± 5.98**	**2.96 ± 1.77 ^3^**	0.996	0.467
Shimmer	2.45 ± 3.38	1.93 ± 4.04	0.95 ± 0.75	0.62 ± 0.29	0.088	0.158
NHR	0.35 ± 0.25	0.22 ± 0.10	0.35 ± 0.34	0.17 ± 0.06	0.965	0.093
GRBAS sum	**8.71 ± 2.36**	**5.30 ± 1.83 ^3^**	**8.00 ± 4.04**	**5.40 ± 3.93 ^3^**	0.587	0.940
Grade	2.42 ± 0.67	1.70 ± 0.48	**2.44 ± 0.86**	**1.60 ± 1.05 ^3^**	0.925	0.777
Roughness	1.63 ± 0.83	1.30 ± 0.48	**1.82 ± 0.95**	**1.35 ± 0.93 ^3^**	0.565	0.875
Breathiness	**2.42 ± 0.79**	**1.10 ± 0.74 ^3^**	**2.29 ± 0.92**	**1.05 ± 1.05 ^3^**	0.712	0.894
Asthenia	1.17 ± 0.94	0.50 ± 0.71	1.35 ± 0.93	0.85 ± 1.04	0.601	0.348
Strain	1.08 ± 1.08	0.70 ± 0.67	0.88 ± 1.05	0.85 ± 1.04	0.621	0.575

Data are expressed as mean ± standard deviation. Abbreviations: VHI, voice handicap index; MPT, max phonation time; sec, seconds; NHR, noise-to-harmonic ratio; Hz, hertz. ^1^
*p* value when the preoperative variable in the age ≥ 60 subgroup was compared with that in the age < 60 subgroup. ^2^
*p* value when the postoperative variable in the age ≥ 60 subgroup was compared with that in the age < 60 subgroup. ^3^
*p* <0.05 when the postoperative variable was compared with that in the preoperative variable. Significant *p* values are marked in bold.

**Table 4 healthcare-08-00521-t004:** Subgroup comparisons of parameters between men and women.

Variable	Men (n = 23)		Women (n = 12)		Men vs. Women	
	Pre-Operation	Post-Operation	Pre-Operation	Post-Operation	*p*-Value ^1^	*p*-Value ^2^
VHI-10	**29.00 ± 4.24**	**4.28 ± 2.30 ^3^**	**31.75 ± 7.80**	**7.00 ± 1.41 ^3^**	0.677	0.494
MPT (sec)	**5.04 ± 4.23**	**10.12 ± 6.15 ^3^**	**4.02 ± 0.88**	**7.99 ± 2.89 ^3^**	0.509	0.362
F0 (Hz)	178.90 ± 82.58	151.78 ± 48.28	**242.93 ± 60.49**	**204.11 ± 50.46 ^3^**	**0.033**	**0.007**
Jitter	**6.61 ± 5.76**	**2.96 ± 2.23 ^3^**	5.81 ± 4.32	3.52 ± 1.63	0.695	0.456
Shimmer	1.66 ± 2.41	1.43 ± 3.15	1.49 ± 2.41	0.58 ± 0.24	0.854	0.363
NHR	0.38 ± 0.33	0.20 ± 0.09	0.30 ± 0.24	0.18 ± 0.07	0.454	0.492
GRBAS sum	**7.98 ± 3.39**	**4.84 ± 3.25 ^3^**	**8.90 ± 3.70**	**6.27 ± 3.44 ^3^**	0.496	0.265
Grade	**2.38 ± 0.86**	**1.53 ± 0.96^3^**	2.56 ± 0.53	1.82 ± 0.75	0.580	0.396
Roughness	1.58 ± 0.91	1.16 ± 0.83	2.11 ± 0.78	1.64 ± 0.67	0.137	0.117
Breathiness	**2.30 ± 0.92**	**1.00 ± 0.94 ^3^**	**2.44 ± 0.73**	**1.18 ± 0.98 ^3^**	0.682	0.620
Asthenia	1.25 ± 1.02	0.63 ± 0.90	1.33 ± 0.71	0.91 ± 1.04	0.826	0.448
Strain	0.75 ± 1.07	0.53 ± 0.70	**1.44 ± 0.88**	**0.73 ± 0.65 ^3^**	0.101	0.441

Data are expressed as mean ± standard deviation. Abbreviations: VHI, voice handicap index; MPT, max phonation time; sec, seconds; NHR, noise-to-harmonic ratio; Hz, hertz. ^1^
*p* value when the preoperative variable in the men subgroup was compared with that in the women subgroup. ^2^
*p* value when the postoperative variable in the men subgroup was compared with that in the women subgroup. ^3^
*p* < 0.05 when the preoperative variable was compared with that in the postoperative variable. Significant *p* values are marked in bold.

**Table 5 healthcare-08-00521-t005:** Subgroup comparisons of parameters between the BMI ≥ 24 and BMI < 24 subgroups.

Variable	BMI ≥ 24 (n = 16)		BMI < 24 (n = 15)		High BMI vs. Low BMI	
	Pre-Operation	Post-Operation	Pre-Operation	Post-Operation	*p*-Value ^1^	*p*-Value ^2^
VHI-10	**32.75 ± 6.24**	**7.33 ± 1.15 ^3^**	**28.43 ± 5.32**	**9.33 ± 4.61 ^3^**	0.857	0.703
MPT (sec)	**4.58 ± 3.69**	**9.36 ± 6.74 ^3^**	**5.20 ± 3.71**	**9.60 ± 4.73 ^3^**	0.686	0.922
F0 (Hz)	**185.15 ± 61.71**	**175.00 ± 57.50 ^3^**	208.34 ± 93.32	159.26 ± 56.51	0.450	0.473
Jitter	**6.54 ± 4.73**	**2.95 ± 1.50 ^3^**	5.57 ± 5.01	3.43 ± 2.59	0.608	0.564
Shimmer	0.93 ± 0.48	0.63 ± 0.35	1.83 ± 2.93	1.65 ± 3.63	0.268	0.325
NHR	0.35 ± 0.25	0.19 ± 0.09	0.34 ± 0.34	0.20 ± 0.08	0.949	0.873
GRBAS sum	**9.11 ± 3.29**	**6.15 ± 3.31 ^3^**	**7.33 ± 3.66**	**4.36 ± 3.50 ^3^**	0.182	0.184
Grade	**2.50 ± 0.76**	**1.69 ± 0.75 ^3^**	**2.36 ± 0.84**	**1.43 ± 1.02 ^3^**	0.641	0.453
Roughness	1.68 ± 1.10	1.46 ± 0.88	**1.85 ± 0.69**	**1.14 ± 0.77 ^3^**	0.643	0.325
Breathiness	**2.43 ± 0.76**	**1.23 ± 0.93 ^3^**	**2.23 ± 1.01**	**0.86 ± 1.03 ^3^**	0.568	0.332
Asthenia	1.36 ± 0.74	0.92 ± 0.95	1.08 ± 1.04	0.57 ± 1.02	0.425	0.364
Strain	1.14 ± 1.10	0.85 ± 0.69	0.77 ± 1.01	0.36 ± 0.63	0.368	0.066

Data are expressed as mean ± standard deviation. Abbreviations: VHI, voice handicap index; MPT, max phonation time; sec, seconds; NHR, noise-to-harmonic ratio; Hz, hertz. ^1^
*p* value when the preoperative variable in the BMI ≥ 24 subgroup was compared with that in the BMI < 24 subgroup. ^2^
*p* value when the postoperative variable in the BMI ≥ 24 subgroup was compared with that in the BMI < 24 subgroup. ^3^
*p* < 0.05 when the postoperative variable was compared with that in the preoperative variable. Significant *p* values are marked in bold.

**Table 6 healthcare-08-00521-t006:** Repeated-measures ANOVA for analysis of different effect on variables.

Variable	Operation Effect (Pre- vs. Post- op)		Age Effect (Age vs. Variable)		BMI Effect (BMI vs. Variable)		Sex Effect (Sex vs. Variable)	
	F	*p*	F	*p*	F	*p*	F	*p*
VHI-10	29.909	**0.012**	3.528	0.201	1.728	0.319	3.528	0.201
MPT (sec)	6.152	**0.020**	0.006	0.940	0.026	0.873	0.003	0.956
F0 (Hz)	8.017	**0.009**	0.031	0.861	0.568	0.458	0.241	0.627
Jitter	7.097	**0.013**	0.217	0.645	0.602	0.445	0.152	0.700
Shimmer	1.476	0.235	0.167	0.686	1.110	0.302	1.390	0.249
NHR	4.694	**0.039**	0.544	0.467	0.029	0.865	0.018	0.893
GRBAS sum	18.761	**<0.001**	0.007	0.933	0.439	0.514	0.137	0.715
Grade	13.848	**0.001**	0.280	0.602	1.223	0.280	0.065	0.800
Roughness	5.297	**0.030**	0.006	0.941	0.002	0.964	0.037	0.848
Breathiness	20.911	**<0.001**	0.478	0.496	0.337	0.567	0.111	0.742
Asthenia	5.233	**0.031**	0.531	0.473	0.018	0.895	5.836	0.024
Strain	4.239	**0.05**	0.076	0.785	0.012	0.914	0.060	0.809

Data are expressed as mean ± standard deviation. Abbreviations: VHI, voice handicap index; MPT, max phonation time; sec, seconds; NHR, noise-to-harmonic ratio; Hz, hertz. Significant *p* values are marked in **bold**.

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
