# Peer review of "Patient-Related Factors of Medialization Laryngoplasty with Autologous Thyroid Cartilage"

_healthcare, 2020, doi:10.3390/healthcare8040521_

Round 1
Reviewer 1 Report
Abstract, Introduction, Material and methods sections are well written and clear.
In the procedure chapter, it is not explained if the cartilage position was adjusted regarding the voice obtained during surgery.
12 months follow up is not a very long time in denervated larynx.
In the results chapter: figure 1 and Table I are not very different.
Discussion: no comparison with the other available techniques
Reviewer 2 Report
Sehr geehrte Autoren,
die vorliegende retrospektive Auswertung hat eine relevante Bedeutung für die klinische Patientenversorgung. Die 3 Hauptaspekte Alter, Geschlecht und Körpergewicht sind interessant. Wurde die erzielte Stimmfunktion intraoperativ überprüft und die OP angepasst? Wie hoch war die Komplikationsrate und welche Komplikationen traten auf? Die Tabelle 1 ist unübersichtlich. Aufgrund der Probandenanzahl von 35 wäre auch eine ausführliche Darstellung der Daten sinnvoll.
Die Statistik ist unklar. Wäre eine Korrektur der multiplen Testung z.B. mit Bonferoni für alle Untertests notwendig? Nach welchem Zeitraum wurden die postoperativen Werte in Tabelle 2 bis 5 bestimmt.
In Figure 3 sind die Grafiken b,d,f nicht ausreichend erläutert. Wird die MPT mit der cummulative voice recovery gleichgesetzt?
Die Diskussion sollten auch die Ergebnisse nach Medialisation mit anderen Materialien wie Silikonblock oder Friedrichimplantat diskutiert werden.
Insgesamt ist es eine wertvolle Arbeit mit Relevanz für die klinische Versorgung.
Reviewer 3 Report
This report of longitudinal voice data from a retrospective cohort of patients undergoing MLATC is a welcomed addition to the literature. The fact that only 35 patients were available from a 7-year period indicates that this procedure is rare. Voice, and importantly, voice satisfaction (per the VHI-10), improved post-operatively overall.
In this review, I will point out some issues that, if addressed, will improve the paper.
Abstract. Please reword "men and females" to "men and women"
Background
1. Did the surgical procedure or frequency of these procedures change over the 7-year recruitment period?
Method and Results
- The VHI-10 is not an acoustic variable. It is a self-report questionnaire.
- The rationale for the age cut-off is not relevant. Enough is known about the effect of aging on voice to based the cut-off value on voice changes. Better yet, do not group participants based on two age groups; age can be included as a continuous variable in a multiple regression model or as a covariate in a repeated-measures ANOVA.
- The same issue as that for age pertains to BMI.
- Was the MPT task used for all acoustic analyses? If not, what task was used? If so, please justify. This is not a natural task and its purpose is not to represent typical vocalization. It is clear from the F0 data in table 4 that the task was not natural (e.g., mean F0 for men was 179 Hz pre-op -- this is high for vocally normal men (normal range is approximately 100-140 Hz), and is especially high for men with UVFP).
- Did the authors consider using cepstral analysis of phonation? This yields better acoustic correlates to dysphonia, and breathiness in particular, than jitter, shimmer, and NHR.
- Please provide more information about the double-blinding and perceptual rating procedures by the 2 raters. How were the acoustic samples presented? Were the raters together? Were ratings averaged? If they were decided by consensus, how were disagreements resolved? What was the percentage agreement between the raters?
- Using multiple t-tests is not the best way to analyze these data. Repeated-measures ANOVA would be a better option. Importantly, age and weight can be included as continuous variables. It would make more sense to analyze the data statistically with these as design variables in a single model rather than analyzing each factor separately as binary variables. This will also help eliminate the large amount of redundant information (including in tables and figures) in the results. If multiple t-tests are used, there should be correction factors included.
- The longitudinal aspect of this work is interesting, but the presentation of these results was confusing. Sometimes, just pre- and post-op was presented. At other times, monthly data were presented. This reviewer had difficulty following the redundancy in reporting the data in various formats and iterations.
Discussion and Conclusion
- I think that the conclusions will change when the data are analyzed using different statistical models, and this will affect the discussion substantially.
- Why would patients prefer this more invasive surgical approach to injection thyroplasty?
Thank you for the opportunity to review this paper.
